# Circulating Dopamine Is Regulated by Dietary Glucose and Controls Glucagon-like 1 Peptide Action in White Adipose Tissue

**DOI:** 10.3390/ijms24032464

**Published:** 2023-01-27

**Authors:** Gabriela Tavares, Daniela Rosendo-Silva, Flávia Simões, Hans Eickhoff, Daniela Marques, Joana F. Sacramento, Adriana M. Capucho, Raquel Seiça, Sílvia V. Conde, Paulo Matafome

**Affiliations:** 1Institute of Physiology and Institute of Clinical and Biomedical Research (iCBR), Faculty of Medicine, University of Coimbra, 3000-548 Coimbra, Portugal; 2Center for Innovative Biomedicine and Biotechnology (CIBB), University of Coimbra, 3000-548 Coimbra, Portugal; 3Clinical-Academic Center of Coimbra, 3004-531 Coimbra, Portugal; 4NOVA Medical School, Faculdade de Ciências Médicas, Universidade Nova de Lisboa, 1169-056 Lisbon, Portugal; 5Instituto Politécnico de Coimbra, Coimbra Health School, 3046-854 Coimbra, Portugal

**Keywords:** dopamine, GLP-1, white adipose tissue

## Abstract

Dopamine directly acts in the liver and white adipose tissue (WAT) to regulate insulin signaling, glucose uptake, and catabolic activity. Given that dopamine is secreted by the gut and regulates insulin secretion in the pancreas, we aimed to determine its regulation by nutritional cues and its role in regulating glucagon-like peptide 1 (GLP-1) action in WAT. Solutions with different nutrients were administered to Wistar rats and postprandial dopamine levels showed elevations following a mixed meal and glucose intake. In high-fat diet-fed diabetic Goto-Kakizaki rats, sleeve gastrectomy upregulated dopaminergic machinery, showing the role of the gut in dopamine signaling in WAT. Bromocriptine treatment in the same model increased GLP-1R in WAT, showing the role of dopamine in regulating GLP-1R. By contrast, treatment with the GLP-1 receptor agonist Liraglutide had no impact on dopamine receptors. GLP-1 and dopamine crosstalk was shown in rat WAT explants, since dopamine upregulated GLP-1-induced AMPK activity in mesenteric WAT in the presence of the D2R and D3R inhibitor Domperidone. In human WAT, dopamine receptor 1 (*D1DR*) and *GLP-1R* expression were correlated. Our results point out a dietary and gut regulation of plasma dopamine, acting in the WAT to regulate GLP-1 action. Together with the known dopamine action in the pancreas, such results may identify new therapeutic opportunities to improve metabolic control in metabolic disorders.

## 1. Introduction

Besides its involvement in the regulation of several neuronal pathways in the CNS, dopamine has also been described to regulate several processes in the periphery, such as gut motility, kidney salt excretion, the chemosensory reflex of the carotid body, and the regulation of insulin secretion by β-pancreatic cells [1,2,3,4,5,6]. Interestingly, Chaudhry and co-workers have proposed that dopamine may act as an anti-incretin in pancreatic β-cells through activation of dopamine type 2 receptor (D2R), as it inhibits glucose- and GLP-1-stimulated insulin secretion [2]. Such inhibition was also observed for the D2R agonist, bromocriptine, suggesting that modulation of D2R could be a mechanism protecting from beta-cell exhaustion [4,6]. However, bromocriptine is also an agonist of the D3R, as well as of the 5-hydroxytryptamine receptors 1A, 2A, 1D and 1C, and of the α-2A-adrenergic receptor [7]. Peripheral dopamine production has been attributed to several sources, such as the adrenal glands and the sympathetic nerve terminals, and the gut, especially the pancreas [2,8,9]. Indeed, dopamine biosynthesis machinery has been described all along the gastrointestinal tract, being therefore suggested as one of the main sources of circulating dopamine [10]. This catecholamine has been shown to regulate gut motility and to be regulated by nutritional signals since its plasma concentrations were recently demonstrated to rise after consuming a mixed meal [2,7]. However, the mechanisms for dopamine regulation after a meal, the nutritional signals involved, and their role in regulating the postprandial metabolic processes are still not entirely known.

Our group has recently shown that dopamine directly modulates glucose uptake independently of insulin signaling in the skeletal muscle (through a D1R mechanism) and in the liver (through a D2R mechanism) [11]. In WAT, dopamine enhanced insulin-mediated glucose uptake also via D2R [11]. Moreover, dopamine-activated catabolic pathways in the WAT in the presence of a D2R antagonist suggest distinct and complementary activities in controlling glucose uptake and catabolic activity in the post-prandial and postabsorptive periods [12]. Accordingly, bromocriptine was observed to upregulate insulin signaling in the adipose tissue of high-caloric diet-fed diabetic rats, also promoting the activation of catabolic pathways in the postabsorptive phase [11,12]. Thus, peripheral dopamine emerges as a regulator of insulin signaling and metabolic activity in the adipose tissue. However, the crosstalk with other mechanisms involved in postprandial glucose metabolism, such as incretins, is unknown. Given the role of dopamine in regulating insulin secretion by the pancreas and sensitivity in the adipose tissue, we hypothesized that it might also modulate the action of GLP-1 in such tissue. Thus, the aim of the present study is to determine the postprandial regulation of plasma dopamine levels by nutritional cues and its crosstalk with GLP-1 action in adipose tissue toward the regulation of glucose and lipid metabolism.

## 2. Results

### 2.1. The Gut Is Involved in Post-Prandial Dopamine Excursions and Peripheral Signaling

It has been described that plasma dopamine levels increase following ingestion of a mixed meal, in both humans and rats [2,10]. The composition of such diets usually consists of a load of 50% carbohydrates, 30% fat, and 20% protein. To determine the contribution of single macronutrients on the nutritional modulation of dopamine secretion we fed fasted 10-week-old Wistar rats with either glucose, starch, arginine, corn oil solutions, or bovine serum albumin solution, according to the prevalence of each nutrient in the mixed meal. As expected, mixed meal ingestion (blue) induced an increase in plasma dopamine levels 30′ after gavage (Figure 1C, *p* < 0.05), which was not observed for the vehicle or sham procedure (Figure 1A,B). Carbohydrate consumption elicited dopamine excursions, with glucose (60 mg/kg) (red) being the most effective, with a post-prandial dopamine peak 15′ after gavage (Figure 1D,G, *p* < 0.05 vs. Vehicle and Sham). Intriguingly, high-concentration glucose solutions (red) showed less robust effects on plasma dopamine levels, with a non-significant increase with 200 mg/kg at 30′ and no effect with the higher dosage (500 mg/kg) (Figure 1H). Starch feeding (orange) tendentially elicited dopamine in circulation at the lowest dosage (246 mg/kg) at 30 min after gavage, despite not reaching statistical significance (Figure 1E,I), with no effects being observed for the highest dosage of 500 mg/kg (Figure 1J). Regarding corn oil, albumin, and arginine, no dopamine fluctuations were detected after gavage when compared to both sham and vehicle groups (Figure 1K).

Close to half of the dopamine levels in the organism are suspected to come from the gut since TH expression and activity are robust in mesenteric organs [13]. To disclose the gut’s involvement in postprandial dopamine secretion and action in adipose tissue, we performed sleeve gastrectomy, a restrictive surgical procedure that alters gastric chyme flow, in obese type 2 diabetic GK rats (Figure 2A). Unexpectedly, we found that TH levels were not altered in GKHCD_SL rats (Figure 2B), and neither were plasma dopamine levels upon mixed meal ingestion (Figure 2D,H). At 6-months-old, Wistar rats presented plasma dopamine fluctuation throughout time after a mixed meal consumption (Figure 2E, *p* < 0.01 between baseline and 15′ and *p* < 0.05 between baseline and 30′), which was not observed in lean and obese GK rats (Figure 2C, *p* < 0.05 vs. WSD at 15′, 30′ and 60; Figure 2F,G vs. own baseline). In opposition, no alterations in dopamine postprandial excursions were found in GKHCD_SL in comparison with HCD rats (Figure 2D) and its own baseline (Figure 2H). Given the high responsiveness of peripheral organs to gut nutrient sensing, we then investigated dopaminergic machinery in peripheral regulators of metabolism, the adipose tissue, and the liver. We found that dopamine signaling machinery was deeply restructured following gut remodeling through sleeve gastrectomy, especially in the pEWAT, with an increase in both D1R (Figure 2J, *p* < 0.01 vs. WSD and *p* < 0.05 vs. GKSD and vs. GKHCD) and D2R (Figure 2K, *p* < 0.001 vs. GKSD and *p* < 0.05 vs. GKHCD and GKHCD_Sh), TH (Figure 2L, *p* < 0.01 vs. GKHCD and GKHCD_Sh) and DARPP32 levels, the downstream signal of D1R activation (Figure 2M, *p* < 0.05 vs. GKSD and GKHCD_Sh). No significant alterations were found in dopamine receptors nor DARPP32 levels in the liver of sleeve gastrectomy-submitted rats (Figure 2N,O,Q), while TH levels were increased (Figure 2P, *p* < 0.05 vs. WSD).

### 2.2. Dopamine and GLP-1 Machinery Are Correlated in Human Visceral Adipose Tissue

Recently, Maffei et al. described that dopamine acts as an anti-incretin, counteracting GLP-1 action by triggering decreased glucose-stimulated insulin secretion from β-cells [6]. Herein, we investigated the relationship between GLP-1 and dopamine machinery in human visceral adipose tissue of subjects with obesity and at several degrees of metabolic dysregulation (Figure 3A). We have previously shown that both *DRD1* and *DRD4* were downregulated in the visceral adipose tissue of insulin-resistant obese patients, being correlated with the lower expression of the insulin receptor and lipid oxidation markers [12]. In opposition, *GLP1R* expression in the adipose tissue is not significantly altered with insulin resistance, prediabetes, or type 2 diabetes onset, in the human visceral adipose tissue (Figure 3B). Nevertheless, *GLP1R* expression was observed to be positively correlated with both *DRD1* and *DRD4* (Figure 3C, r = 0.343, *p* = 0.0012 and Figure 3E, r = 0.320, *p* = 0.0026), showing no associations with *D2DR* (Figure 3D). Such moderate positive correlations are maintained and increased only in patients with insulin resistance in relation to *DRD1* (Figure 3I, r = 0.386, *p* = 0.0008) and *DRD4* (Figure 3K, r = 0.419, *p* = 0.0003), being absent in insulin-sensitive patients (Figure 3F–H). Such results suggest that the downregulation of dopamine receptors observed in patients with insulin resistance is correlated with lower *GLP1R* expression.

### 2.3. Dopamine Receptors Differentially Modulate Dopamine-GLP-1 Crosstalk to Modulate AMPK-Thr172 Phosphorylation in the Adipose Tissue

The crosstalk between GLP-1 and dopamine on the regulation of energy metabolism was investigated by assaying AMP-activated protein kinase (AMPK)-Thr172 phosphorylation levels in mesenteric (mWAT), and pEWAT, incubated with GLP-1R agonist liraglutide and/or dopamine, combined with either haloperidol (antagonist for both D1R and D2R) or domperidone (D2R and D3R antagonist) (Figure 4A). In mWAT, dopamine had no effect on inducing AMPK-Thr172 phosphorylation, which is also in accordance with our previous results [11]. Likewise, neither liraglutide alone nor when in combination with dopamine showed any effects (Figure 4B). Haloperidol + liraglutide showed no effect on AMPK activation independently of co-incubation with dopamine (Figure 4B). Co-incubation of dopamine + liraglutide + domperidone significantly enhanced AMPK-Thr172 levels (Figure 4B, *p* < 0.01 vs. CTL, *p* < 0.05 vs. dopamine + Liraglutide). Domperidone coupled with liraglutide abrogated this effect (Figure 4B, *p* < 0.01 vs. dopamine + liraglutide + domperidone), suggesting that D2R inhibition triggers a synergic effect of dopamine and liraglutide on increasing AMPK-Thr172, that is highly dependent on dopamine action on other receptors, as the effect is lost on dopamine absence.

In pEWAT adipose tissue, GLP-1 alone increased AMPK-Thr172 (Figure 4C, *p* = 0.0509 vs. CTL). Haloperidol showed no effect on the stimulatory effect of liraglutide on AMPK activation (Figure 4C). Dopamine co-incubation with liraglutide and domperidone had no significant effect on AMPK-Thr172, but D2R antagonism coupled with liraglutide curbed the AMPK-Thr172 (Figure 4C, *p* < 0.01, vs. liraglutide, *p* < 0.05, vs. dopamine + liraglutide + domperidone), showing the relevance of dopamine action. The levels of Insulin receptor-Tyr972 (InsR-Tyr972) were unchanged under all tested conditions both in mWAT and pEWAT (Figure 4D and 4E, respectively).

### 2.4. Bromocriptine Increases GLP-1R in Adipose Tissue, while Liraglutide Fails to Modulate Dopamine Receptors

To further explore the long-term crosstalk between peripheral signaling of these two metabolic mediators, dopamine signaling was modulated through bromocriptine administration in GKHCD rats (Figure 5A). We found a direct action of bromocriptine in increasing GLP-1R signaling, exclusively in adipose tissue depots. In the pEWAT, GLP-1R is decreased in GKSD rats (Figure 5B, *p* < 0.001 vs. WSD) and in obese type 2 diabetic animals, (Figure 5B, *p* < 0.01 GKHCD and GKHCD_Vh vs. WSD). Bromocriptine treatment significantly improved GLP-1R levels in pEWAT to those similar to the healthy WSD group (Figure 5B, *p* < 0.001 vs. GKSD, and *p* < 0.01 vs. GKHCD and GKHCD_Vh). Brown adipose tissue (BAT) GLP-1R levels were diminished in lean and obese type 2 diabetic animals compared to healthy WSD (Figure 4C, *p* < 0.05 vs. GKSD and *p* < 0.01 vs. GKHCD and GKHCD_Vh) which were restored upon bromocriptine administration (Figure 5C, *p* < 0.05 vs. GKHCD and vs. GKHCD_Vh). In other peripheral organs with a high impact on overall metabolic status, such as the liver and skeletal muscle, GLP-1R levels were not altered after bromocriptine treatment (Figure 5D,E), corroborating previous results of ours suggesting a specific modulatory effect of dopamine in adipose depots 9. Hypothalamic and pancreatic GLP-1R levels were, also, unchanged in bromocriptine-treated rats (Figure 5F,G). Moreover, no alterations related to TH levels were found among groups in the pancreas (Figure 5H).

To understand the role of dopamine in GLP-1 secretion, the levels of GLP-1 in the intestinal wall of Bromocriptine-treated rats were determined. Although a trend to increase GLP-1 reactivity in the ileum wall both by Western Blot and fluorescent immunohistochemistry, no significant results were obtained for GLP-1 and DPP-IV (Appendix A). Accordingly, no significant changes in postprandial plasma GLP-1 levels were observed (Appendix A). Thus, although Bromocriptine significantly upregulated adipose tissue GLP-1R levels, the effects on GLP-1 secretion were not evident.

To better disclose the dopamine-GLP-1 crosstalk, we investigated the effect of liraglutide, a well-known GLP-1R agonist, in the WAT and liver dopaminergic signaling (Figure 6A). Interestingly, liraglutide treatment did not alter dopamine receptors’ levels in the pEWAT, nor influenced dopamine biosynthesis through TH activation (Figure 6B–E). Similarly, in the liver of Liraglutide-treated animals, both D1R and D2R levels were not altered, as well as TH and DARPP32 levels (Figure 6F–I), suggesting that this crosstalk has unilateral effects played by dopamine to enhance GLP-1 peripheral action on adipose depots.

## 3. Discussion

In this study, we demonstrate that plasma dopamine levels are postprandially elicited by dietary glucose, and its action in WAT is regulated by the gut, as demonstrated in rats submitted to sleeve gastrectomy. Previous data have shown the regulation of adipose tissue insulin signaling by dopamine [12]. Here we show that dopaminergic signaling is correlated to GLP-1R in human adipose tissue, their signaling crosstalk to regulate catabolic activity, and dopamine regulates GLP-1R levels in rat adipose tissue, but not the opposite (Figure 7).

Peripheral dopamine has a role in blood pressure, gastrointestinal motility, glucose homeostasis, and body weight control [3,13,14,15]. Initially thought to be exclusively produced at the level of the sympathetic nerves, pancreas, or adrenal medulla, dopamine in peripheral circulation is nowadays also acknowledged to be secreted by the gut [2,10,13]. However, the stimuli underlying intestinal dopamine release are quite unknown. Goldstein et al. have shown that mixed meal consumption by healthy individuals increases plasma dopamine levels, especially dopamine sulfate [16], an effect that we and others have confirmed also in rats [6]. We wondered which nutrient(s) present in the mixed meal could be eliciting dopamine secretion. When challenging Wistar rats with different gavage-administered solutions, rich in each macronutrient, we found that glucose (60 mg/kg), at the same dose present in the mixed meal, has a positive effect in triggering plasma dopamine increases (50% increase). The observed plasma dopamine peak (around 1.5 nmol/L) is consistent with Maffei et al. report using mixed meal-gavage Sprague-Dawley rats [6]. Intriguingly, higher doses of glucose (supraphysiologic)—200 mg/kg and 500 mg/kg—fail to induce dopamine release, suggesting that dopamine plays an important role in the first phase gut-based layer of glucose homeostasis, responding only to lower intakes of glucose. Gut-derived serotonin is secreted upon glucose luminal sensing, but only at higher concentrations (stimulation of enterochromaffin cells with 500 mmol/L of glucose) [17], suggesting an interesting interplay between dopamine and serotonin on the response to luminal glucose cues. The effect of starch feeding on mildly increasing dopamine levels at about 30 and 45 min after gavage, despite not being statistically significant, suggests that its breakage into glucose would eventually result in even higher dopamine levels later in time. The mechanisms behind glucose-stimulated dopamine secretion are still to be uncovered. Recently, a study has shown that intragastric sucrose administration led to a vagus-mediated activation of dopaminergic neurons in the ventral tegmental area [18]. Together, this evidence supports dopamine as a sensor for physiological post-ingestive glucose, informing on this nutritional signal in central and peripheral energy balance governors.

There is no doubt that non-neuronal dopamine production happens within the gastrointestinal tract, despite its precise location is still unknown. TH, the enzyme that originates dopamine precursor, was shown to be expressed in the ileum and the oxyntic stomach [2,13,19]. In addition, D1R, dopamine transporter, and both vesicular monoamine transporters 1 and 2 were found in the stomach’s parietal and enterochromaffin cells [2,19]. Measurement of dopamine content in stomach tissue was not different in control versus chemical sympathectomy-submitted rats, nor was TH activity, meaning that non-neuronal peripheric dopamine production takes place in the stomach [19]. To further explore the involvement of the gut on dopamine secretion and action in peripheral tissues, we performed the restrictive (leaving only 20% of the stomach’s volume) metabolic surgery method vertical sleeve gastrectomy in HCD-fed GK rats (GKHCD_SL), accelerating gastric emptying and leading to marked alterations on nutrient sensing and gut hormones secretion profile [19,20]. After challenging GKHCD_SL rats with a mixed meal intragastric injection, we found that plasma dopamine levels were surprisingly unaltered. Apparently, GK rats were insensitive to the post-prandial dopamine secretion profile seen in healthy Wistar rats, independently of the obesity condition, and sleeve gastrectomy fails to rescue such a secretion profile. In fact, in patients with type 2 diabetes, the levels of circulating tyrosine are decreased relative to healthy controls [21]. In fact, dopamine secretion was unresponsive to sleeve gastrectomy suggesting that dopamine production might not be entirely attributed to the gut/stomach, or at least, its’ modulation is not similar to incretins such as GLP-1. This assumption is corroborated by the unaltered TH ileum levels upon sleeve gastrectomy. However, we showed that sleeve gastrectomy upregulates WAT dopaminergic machinery and TH, while no effects were found in the liver, with the exception of TH-increased levels. The mechanisms for such regulation are still vague but may involve neuronal pathways regulated by dopamine or by other gut signals since no plasma postprandial dopamine oscillation was observed under the same conditions.

For years, dopamine has been known as an important regulator of energy homeostasis. Recently, Maffei et al., described dopamine as an anti-incretin, showing evidence of its role in inhibiting insulin release from β-cells, thus counteracting the incretin effect of GLP-1 [6]. Interestingly, dopamine plasma kinetics are congruent to those of the incretin hormone GLP-1, upon mixed meal consumption by healthy individuals [2]. As Chaudry et al. proposed, the coincident kinetics and the opposing effects on glucose-stimulated insulin secretion, suggest a relevant crosstalk of dopamine and GLP-1 on glucose homeostasis regulation [2]. To further highlight that interplay, we investigated dopamine and GLP-1 machinery in human adipose tissue from obese patients with or without insulin resistance and disruption of homeostatic control of glucose. Recent data that we published showed a decrease in *DRD4* and *DRD1* expression with the onset of insulin resistance, while *DRD2* expression remained unchanged with the progression of metabolic dysregulation in obesity [11]. In comparison to the dopaminergic machinery, *GLP1R* is not substantially expressed in the visceral adipose tissue of obese patients and its expression was also not significantly altered in the adipose tissue of these patients with obesity at distinct degrees of dysmetabolism. Conversely, *GLP1R* gene expression and protein levels were described to be increased in the visceral adipose depot in obese patients with a high degree of insulin resistance [22]. Nevertheless, we found moderate positive correlations between *DRD1/DRD4* and *GLP1R*, especially in patients with insulin resistance, suggesting that *GLP1R* expression is associated with the decreased expression of dopamine receptors in patients with insulin resistance. GLP-1 has been linked to an effect on the level of energy expenditure in the visceral adipose tissue [23]. Recent clinical studies combining D2R and GLP-1R agonists in patients with type 2 diabetes [24] and the increasing interest in GLP-1R agonists for Parkinson’s treatment [25] demand the unraveling of the putative crosstalk dopamine-GLP-1, which led us to speculate if GLP-1R could be stimulated by dopamine receptor agonism.

Our results in the mesenteric tissue explants revealed that D2R inhibition augmented the potential of dopamine and liraglutide to induce AMPK-Thr172 phosphorylation, an effect that was dependent on dopamine presence. While D1R is coupled to a Gαs protein, thus enhancing intracellular cAMP, D2R is, on the contrary, associated with a Gαi/o protein [26]. D1R was recently shown to induce lipolysis by increasing hormone-sensitive lipase levels in the 3T3 adipocyte cell line [27]. Thus, this may mean that dopamine is probably accounting for the increase in AMPK-Thr172 via D1R activation, upon domperidone incubation. The fact that dopamine + liraglutide or dopamine alone did not increase pAMPK-Thr172 levels may be due to simultaneous D2R binding, only resulting in D1R activation upon D2R blockage. The postprandial/postabsorptive modulation of dopamine receptors is in fact also unknown, as well as its impact on cell metabolism. Haloperidol had no effect on the levels of AMPK-Thr172, however, one should note that it is a dual antagonist and may not reflect accurate D1R inhibition. Liraglutide seems to play an irrelevant role in mesenteric adipose tissue catabolic activity but as expected, it had a stimulatory effect over AMPK-Thr172 in the pEWAT depot. In this depot, the crosstalk observed in mesenteric tissue is not observable, which is according to our previous results regarding insulin-dopamine crosstalk [11]. Moreover, such differences suggest that, since mesenteric tissue is closer to the intestine, it may act as an initial buffer of intestinal nutrients. In fact, mesenteric AT is anatomically and metabolically different from the epididymal one. The mesenteric depot is known for having a higher triglyceride turnover rate, draining free fatty acids to the hepatic portal vein and exposing the liver to high concentrations of lipids [28].

Considering not only the beneficial effects of D2R agonism in glycemic control [24,29] but also the possible crosstalk between dopaminergic and GLP-1 signaling, we treated GK rats with bromocriptine, the most frequently and clinically used D2R agonist. Bromocriptine treatment increased GLP-1R levels specifically in adipose tissue depots, both white and brown. This is in accordance with previous results we obtained demonstrating that the adipose tissues are the preferential peripheral targets of Bromocriptine action [12]. Nonetheless, our results show that bromocriptine has no effects on the intestinal production of GLP-1, nor does it stimulate its secretion (Appendix A). To better understand the dopamine-GLP-1 crosstalk we then administered the GLP-1R agonist liraglutide to GK rats. The absence of alterations in dopamine signaling machinery, either in pEWAT and liver, both in Wistar and GK rats treated with liraglutide, suggests that dopamine remodels GLP-1R signaling, but the opposite is not true. While suspected to have a role in the modulation of striatal dopamine levels [30], herein we show that stimulation of GLP-1 has no effect on peripheral dopamine signaling.

To the best of our knowledge, this is the first report claiming that plasma levels of dopamine are responsive to a low-dose glucose intake and exerts a unidirectional stimulation of adipose tissues’ GLP-1R pathways. The holistic view is that dopamine is a powerful energy sensor, being triggered by glucose ingestion and acting through unknown neuroendocrine mechanisms in a two-way fashion: in brain centers, to control feeding behavior and positive energy balance and, in the periphery, to prime adipose reservoirs to optimize energy storage and expenditure. Together with the anti-incretin effect of dopamine in β-cells, this might underlie the success of bromocriptine as an anti-diabetic drug, reducing insulin production, thus preventing β-cells exhaustion, while increasing insulin and GLP-1R sensitivity in peripheral insulin-sensitive tissues, especially the adipose depots.

However, our study has some limitations that should be further addressed. Firstly, we can only suspect that dopamine production and secretion upon mixed-meal consumption comes from the gut, as supported by the literature [2,10]. By measuring plasma dopamine levels, we are unable to ensure that dopamine secretion is indeed coming from the gastrointestinal tract, let alone its precise location. Likewise, although we show that type 2 diabetes abrogates mixed meal-induced dopamine secretion, sleeve gastrectomy failed to increase postprandial dopamine secretion, raising the question of whether dopamine action in the adipose tissue occurs through neuronal or endocrine pathways or both. Another missing piece in the puzzle is the putative bromocriptine-induced GLP-1 secretion. Despite supporting the role of bromocriptine in increasing adipose tissue sensitivity to GLP-1, our data reveal no clear effects on plasma and ileum GLP-1 levels after Bromocriptine treatment, suggesting that dopamine does not control GLP-1 secretion, at least via the D2 receptor.

Altogether, our data highlights the value of the peripheral dopaminergic system as a nutrient sensor and a tool to modulate GLP-1 action in adipose tissue. Although the mechanisms remain elusive, we report herein that dopamine is secreted upon glucose intake and its machinery is upregulated in the adipose tissue upon sleeve gastrectomy, supporting dopamine’s involvement in the post-prandial adaptation of peripheral nutrient partitioning. The disclosure of post-prandial dopamine function in both central and peripheral regulation of energy expenditure might provide new insights regarding the pharmacological value of bromocriptine and possibly other dopaminergic modulators, as well as their possible association with GLP-1R agonists. Even though we focused our attention on GLP-1, the putative dopamine-mediated modulation of other gut hormones should not be depreciated, and this should be acknowledged in the future.

## 4. Materials and Methods

### 4.1. Human Study

Patient selection and characterization: A cohort of obese patients aged 25–65 years old (diabetic and non-diabetic) was selected at the obesity surgery appointment at the Hospital Geral de Coimbra (Covões), Centro Hospitalar Universitário de Coimbra (CHUC). All subjects signed informed consent, and the study was approved by the institutional ethics committee (Ethics Committee of the Coimbra University Hospital Center), according to the principles outlined in the Declaration of Helsinki. Exclusion criteria were: active inflammatory and chronic diseases (neurodegenerative diseases or active tumors), previous restrictive (sleeve gastrectomy) or mal-absorptive (gastric bypass or duodenal switch) surgeries, and type 2 diabetes medication other than metformin (GLP-1RA, dipeptidyl peptidase IV (DPPIV) inhibitors or insulin). On the day before surgery, height and body weight were recorded, and fasting blood samples were collected for biochemical analysis. Visceral WAT samples were collected during surgery and kept in liquid nitrogen to be then stored at −80 °C.

A total of 92 obese patients (77 women and 15 men) were divided into groups according to glycemic profile: fasting glucose levels, HbA1c, and Ox-HOMA2IR. Subject characterization resulted in four different groups: 1–insulin-sensitive group (IS) (*n* = 17), composed by individuals that were both IS (Ox-HOMA2IR < 1) and normoglycemic (NG, fasting glucose < 100 mg/dL and HbA1c < 5.7%); 2–insulin resistant (IR, Ox-HOMA2IR > 1) and NG group (*n* = 29), 3–prediabetic group (*n* = 28), that allocated IR patients with fasting glucose levels from 100 to 125mg/dL or HbA1c between 5.7 and 6.4%; 4–T2D group (*n* = 18), constituted by IR subjects diagnosed with T2D (fasting blood glucose > 125 mg/dL or HbA1c > 6.4%).

Adipose tissue RNA extraction: Total RNA was extracted from visceral WAT human biopsies (100 mg) using an RNeasy Lipid Tissue Mini Kit (Qiagen, Hilden Germany). RNA samples were analyzed by NanoDrop One/One spectrophotometer (ThermoFisher, Waltham, MA, USA) at 260 nm to evaluate their concentration. RNA integrity was also analyzed through capillary electrophoresis with an Agilent RNA 6000 Nano Kit and the results were obtained with the Agilent 2100 Bioanalyser (Agilent Technologies, CA, USA). Then samples were quantified by real-time polymerase chain reaction using the high throughput platform biomarker HD system as described before [9]. Forward and reverse primers (50 μM stock) diluted TE buffer were individually prepared. The primers sequences were: DRD1 forward-ACACAATTAACTCCGTTTCC and reverse-GTAGTGTCCCTGTTTGATTG; DRD2 forward-TCCACTAAAGGGCAACTG and reverse- GGAAACTCCCATTAGACTTC; DRD4 forward-CATCTACACTGTCTTCAACG and reverse-ATTAACGTACAAAAGCGCC; GLP1R forward-GAACTCCAACATGAACTACTG and reverse-CAAAGATGAGGAAGTTCACC obtained by Sigma Aldrich, (St. Louis, MO, USA) and reconstituted in water to a final concentration of 100μM. Then data were collected with Data Collection Software and analyzed using Fluidigm^®^ Real Time PCR Analysis v2.1 software. All data were normalized for the reference gene ATCB.

### 4.2. Animal Models

Wistar rats from our breeding colonies (Faculty of Medicine, University of Coimbra, and NOVA Medical School) were kept under standard conditions [25,26]. The experimental protocol was approved by the local Institutional Animal Care and Use Committees (ORBEA UC 04-2015; and ORBEA NMS 20_03_ORBEA) and by Direção Geral de Agricultura e Veterenária (DGAV). All the procedures were performed by licensed users of the Federation of Laboratory Animal Science Associations (FELASA) and in accordance with the European Union Directive for Protection of Vertebrates Used for Experimental and Other Scientific Ends (2010/63/EU).

#### 4.2.1. Experimental Design for Determining Postprandial Plasma Dopamine Levels

Male Wistar rats (10-week-old, 6 h fasting) were used to evaluate the regulation of plasma dopamine levels by the specific nutrients. Animals were randomly allocated into the groups (*n* = 5–6): (1) gavage fed a mixed meal diet (Nutricia, Fortimel, Nestle, Switzerland); (2) gavage fed a glucose solution corresponding to the concentration described in the mixed meal diet (2 g of glucose in 23 g of Nutricia, Fortimel); (3) gavage fed a starch solution (12.3 g of carbohydrates in 23 g of Nutricia, Fortimel); (4) gavage fed a bovine serum solution and (5) a solution with arginine (5 g of proteins in 23 g of Nutricia, Fortimel); (6) gavage fed corn oil solution (3.3 g of lipids in 23 g of Nutricia, Fortimel). The volume of each nutrient solution administered by gavage was 2 mL. A Vehicle group was established through gavage administration of 2 mL of distilled water, and a Sham group through the insertion of only the gavage probe, to exclude the possible stress effect. Blood samples were collected before and 15, 30, and 45 min after gavage, as described before [31,32].

#### 4.2.2. Experimental Design of the Sleeve Gastrectomy Procedure

Wistar (WSD) and type 2 diabetic Goto-Kakizaki rats (GK) rats (*n* = 6) were kept with ad libitum access to water and standard diet (A03, SAFE, France) (WSD and GKSD) (Figure 2A). A group of GK rats (GKHCD; *n* = 18) was fed a high-caloric diet (customized A03 high-caloric diet with 20% fat plus 20% sucrose, SAFE, France), between 1 and 6 months old. At 4-months-old, GKHCD rats were randomly assigned to three groups (N = 6): no intervention (GKHCD), sleeve gastrectomy (GKHCD_SL), and sham surgery (GKHCD_Sh). Vertical sleeve gastrectomy and sham surgery were performed as described previously [31,32,33]. Animals were sacrificed at 6 months old, and WAT and liver were collected for biochemical analyses.

#### 4.2.3. Experimental Design for Bromocriptine Administration

The experimental design for bromocriptine administration was similar to sleeve gastrectomy, namely the WSD and GKSD groups (Figure 5A). A group of GK rats (GKHCD; *n* = 18) was fed the same high-caloric diet between 1 and 6-months-old. At 5 months old, the HCD-fed group was randomly divided into three groups (*n* = 6): the first without further treatment (GKHCD); HCD rats treated with bromocriptine in the last month (GKHCD_Br), and HCD rats treated with a vehicle DMSO solution during the same period (GKHCD_Vh) [12]. Bromocriptine, gently supplied by Generis^®^, (Amadora, Portugal) was diluted 1:4 DMSO/H2O and administered daily by intraperitoneal (i.p.) injection (10 mg/Kg/day) during the last month. In the vehicle group, the same volume (100 μL) of the vehicle 1:4 DMSO/H2O was administered i.p. during the same period.

#### 4.2.4. Experimental Design of the Liraglutide Administration

Male Wistar rats 14-week-old were randomly divided into two groups (*n* = 6): (1) Control with saline injection (W); (2) Wistar rats injected subcutaneously with liraglutide (Victoza, Novo Nordisk) 200 μg/Kg twice a day for 14 days (WL) (Figure 6A). Age-matched male GK rats were divided into two groups (*n* = 6): (3) Saline-injected Control GK rats (GK); (4) GK rats injected subcutaneously with liraglutide 200 μg/Kg twice a day for 14 days (GKL) as described before [34].

### 4.3. WAT Ex-Vivo Incubations

Animals were left in overnight fasting and euthanized with sodium pentobarbital (60 mg/kg, i.p.) in the morning. WAT (WAT: mesenteric, mWAT and peri-epididymal, pEWAT) was collected (50–1500 mg) in ice-cold 20% O_2_/5% CO_2_-equilibrated tyrode solution (in mM: 140 NaCl, 5 KCl, 2 CaCl_2_, 1.1 MgCl_2_, 10 HEPES, and 5.5 glucose, pH 7.40). Then tissue samples were transferred to a 2 mL Eppendorf tube and incubated at 37 °C for 10 min for stabilization. Then tissue samples were incubated with the following experimental conditions (minimum of N = 4 to each condition): (1) dopamine (Medopa, Medinfar, 10 µM); (2) liraglutide (a synthetic analog of GLP-1, Victoza, Novo Nordisk, 100 mnM); (3) dopamine plus liraglutide; (4) dopamine plus liraglutide plus haloperidol (500 nM; Sigma, Madrid, Spain); (5) liraglutide plus haloperidol; (6) dopamine plus liraglutide plus domperidone (50 nM, Sigma, Madrid, Spain) and (7) liraglutide plus domperidone. After 10 min of incubation, samples were frozen in liquid nitrogen to be quantified by Western blot. Dopamine and its receptors (D1R and D2R) antagonists concentrations were chosen as previously described [11,35]. Haloperidol was used in a concentration that blocks both D2R and D1R, while domperidone was used in a 10-fold less concentration allowing the selective blockage of D2R [11,36].

### 4.4. Western Blotting

Tissues were homogenized as previously described and samples were loaded in 8% polyacrylamide gels, separated by SDS-page, and transferred to a PVDF membrane (Advansta, San Jose, CA, USA). Membranes were incubated with the specific primary antibodies overnight at 4 °C (listed below), and then incubated for 2 h at room temperature with secondary antibodies. The secondary antibodies were anti-mouse (GE Healthcare, Chicago, IL, USA, EUA) and anti-rabbit (Bio-Rad, Des Plaines, IL, USA). Membranes were revealed using ECL substrate in a Versadoc system (Bio-Rad, Des Plaines, IL, USA) and analyzed with Image Quant^®^ (Molecular Dynamics, Sunnyvale, CA, USA).

### 4.5. Reagents, ELISA Kits and Antibodies

Salts and organic solvents used in solution preparations were purchased from Fisher Scientific (Leicestershire, UK), Sigma Chemicals (St. Louis, MO, USA, EUA), or Merck (Darmstad, Germany), with the highest grade of purity commercially available. It used antibodies to analyze D1R, D2R, DARPP32 (ab81296, ab85367, and ab40801 respectively, Abcam, UK), and tyrosine hydroxylase (TH), (T1299, Sigma Aldrich, St. Louis, MO, USA) at a 1:1000 dilution. Moreover, antibodies against GLP-1 and GLP-1R and against the phosphorylated form of insulin receptor were used (ab22625, ab218532, and InsR-Tyr972, ab5678 respectively, Abcam, UK) as well, against phosphorylated AMPK form (Thr172, 2535S, Cell Signalling Technology, Danvers, MA, USA, EUA). Calnexin was used as loading control (AB0037, Sicgen, Cantanhede, Portugal). Plasma dopamine levels and plasma GLP-1 levels were assessed through the Dopamine ELISA Kit, (Abnova, Taiwan) and Rat GLP1/Glucagon-like Peptide 1 ELISA Kit, (LifeSpan BioScience, Inc., Washington, DC, USA) respectively.

### 4.6. Statistical Analysis

Assessment of plasma dopamine excursion upon nutrient feeding through time one-way repeated measures ANOVA with Dunn’s post-hoc corrections for multiple comparisons were performed, while between groups a comparison one-way ANOVA with Tukey’s post-hoc corrections for multiple comparisons was performed. To assess sleeve surgery effect on plasma dopamine excursion, one-way repeated measures ANOVA with Dunn’s post-hoc corrections for multiple comparisons were performed to assess differences in plasma dopamine concentration throughout time after a mixed meal feeding. Comparison between groups was assessed by one-way ANOVA with Tukey’s post-hoc corrections for multiple comparisons. A one way-ANOVA test with Tukey’s post-hoc corrections for multiple comparisons was performed to analyze all protein levels in sleeve gastrectomy surgery, bromocriptine, and liraglutide-treated animal models in all tissues. TH level on ilium and explants data were analyzed using the non-parametric Kruskal-Wallis with multiple comparisons test with Dunn’s post-hoc corrections. Data from adipose tissue ex vivo incubations were analyzed by one-way ANOVA with no corrections for multiple comparisons (Fisher’s LSD test). Results were presented as mean ± SEM. Regarding human data analysis, non-parametric tests were performed (sample size < 30/group), and results were presented as median and interquartile range. The Kruskal-Wallis test was applied to compare GLP-1R gene expression between groups. The Spearman correlation test was performed to assess the correlation between GLP-1 and dopamine receptors gene expression. Differences were considered significant at *p* < 0.05, and all computation analysis was performed using Graphpad Prism (6.0 version, GraphPad Software, Inc., San Diego, CA, USA).

## Figures and Tables

**Figure 1 ijms-24-02464-f001:**
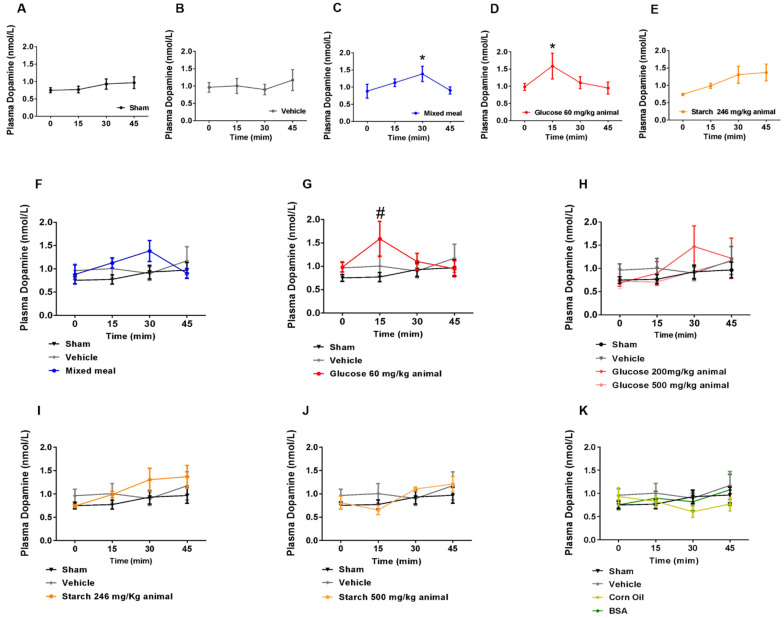
Postprandial dopamine excursions are modulated by gut carbohydrate sensing. Plasma dopamine excursions increased 30 min after mixed meal ingestion (**C**) and 15 min after glucose ingestion (60 mg/kg animal, (**D**)), but not in sham (**A**) and vehicle (**B**) rats. Curves are compared in (**G**). Starch ingestion (246 mg/kg) triggered an increase in plasma dopamine level at 30 min which was maintained in time (45 min, (**E**)), despite lacking a significant statistical result (**E**,**I**). Higher doses of ingested glucose (200 mg/kg or 500 mg/Kg) did not alter plasma dopamine excursions (**H**) nor did higher doses of starch ((**J**), 500 mg/Kg). Albumin, arginine, and oil did not elicit plasma dopamine fluctuations (**K**). Bars represent means ± SEM. Differences between groups were assessed through the One-way ANOVA repeated measures test with Tukey multiple comparisons were used to analyze plasma dopamine levels throughout time (data plots (**A**) to (**E**)). One-way ANOVA test with Tukey multiple comparisons was used to analyze plasma dopamine levels between groups at each time point (data plots (**F**) to (**K**)). * Differences in time (data plots (**A**) to (**E**)); # different from Sham and Vehicle groups. Levels of significance: 1 symbol, *p* < 0.05; 2 symbols, *p* < 0.01; 3 symbols, *p* < 0.001.

**Figure 2 ijms-24-02464-f002:**
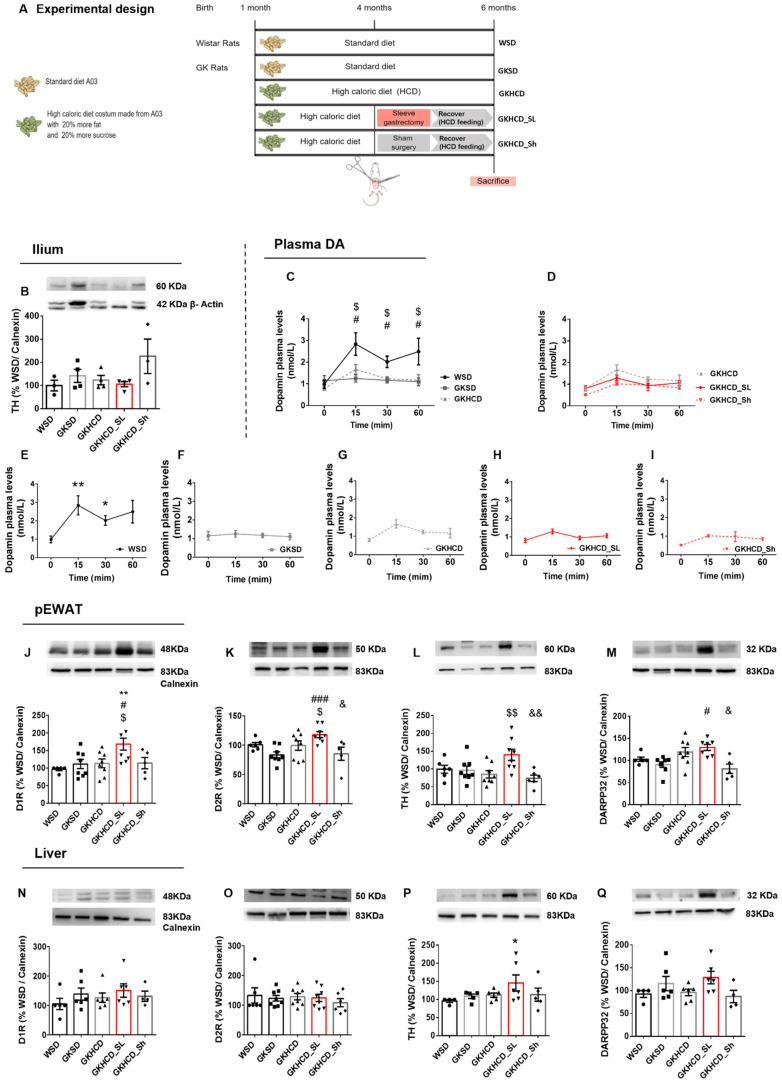
Gut remodeling improved dopaminergic signaling in adipose tissue of animals submitted to sleeve gastrectomy. Vertical sleeve gastrectomy (experimental design depicted in (**A**)) did not alter TH levels in the ilium between groups (**B**). Type 2 diabetic animals fed a standard (GKSD) and a high caloric diet (GKHCD) lost plasma dopamine fluctuations when compared with Wistar control animals (**C**,**E**,**F**,**G**) and gut remodeling with sleeve surgery did not revert this effect (**D**,**G**,**H**,**I**). However, in pEWAT, dopaminergic signaling machinery (D1R (**J**), D2R (**K**), TH (**L**), and DARPP32 (**M**)) was increased in diabetic obese animals after sleeve gastrectomy (GKHCD_SL). The same was not observed in the liver, where D1R (**N**), D2R (**O**), and DARPP32 (**Q**) remained similar between groups, despite TH levels increasing in GKHCD_SL (**P**). Kruskal Wallis test was used to assess TH differences between groups in the ilium (**B**). One-way ANOVA test with Tukey post-hoc corrections for multiple comparisons analysis was used to assess plasma dopamine between groups (data plots (**C**) and (**D**)). To assess plasma dopamine levels throughout time one-way repeated measures ANOVA test with Dunn’s post-hoc correction for multiple comparisons analysis was used (data plots (**E**–**I**)). Bars represent mean ± SEM where symbols represented: # different from GKSD, $ different from GKHCD, and * differences in time. To assess DA signaling machinery in pEWAT and liver tissues and compare between groups, a one-way ANOVA test with Tukey post-hoc corrections for multiple comparisons was used (data plots (**J**–**Q**)). Bars represent means ± SD. Symbols represented * Different from WSD; # different from GKSD; $ different from GKHCD; & different from GKHCD_SL. Level of significance: 1 symbol, *p* < 0.05; 2 symbol, *p* < 0.01; 3 symbols, *p* < 0.001.

**Figure 3 ijms-24-02464-f003:**
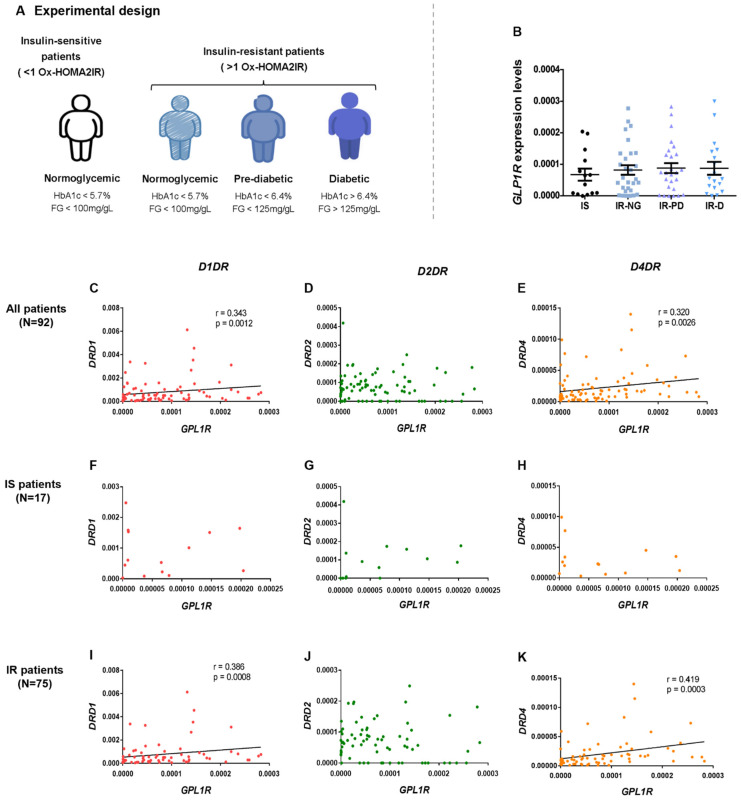
GLP1-R expression in patients with metabolic dysfunction and its correlation with dopamine receptors expression. Expression of *GLP1R* did not alter between groups of obese patients with different grades for glucose and insulin dysmetabolism (**B**). *GLP1R* expression correlates with decreased expression of *DRD1* and *DRD4* in all groups of insulin resistance patients ((**C**,**I**); (**E**,**K**)) but not in insulin-sensitive patients ((**F**,**H**)). *GLP1R* expression did not correlate with *DRD2* for all groups of patients (**D**,**G**,**J**). Bars represent the median ± interquartile range. Kruskal-Wallis was used to assess differences between groups (IS: insulin-sensitivity normoglycemic, IR-NG: Insulin resistance normoglycemic, IR-PD: Insulin resistance pre-diabetic, and IR-D: Insulin resistance diabetic patients). Spearman correlations were used to assess the expression of *GLP1R* correlation with dopamine receptors expression. Levels of significance: 1 symbol, *p* < 0.05; 2 symbols, *p* < 0.01; 3 symbols, *p* < 0.001.

**Figure 4 ijms-24-02464-f004:**
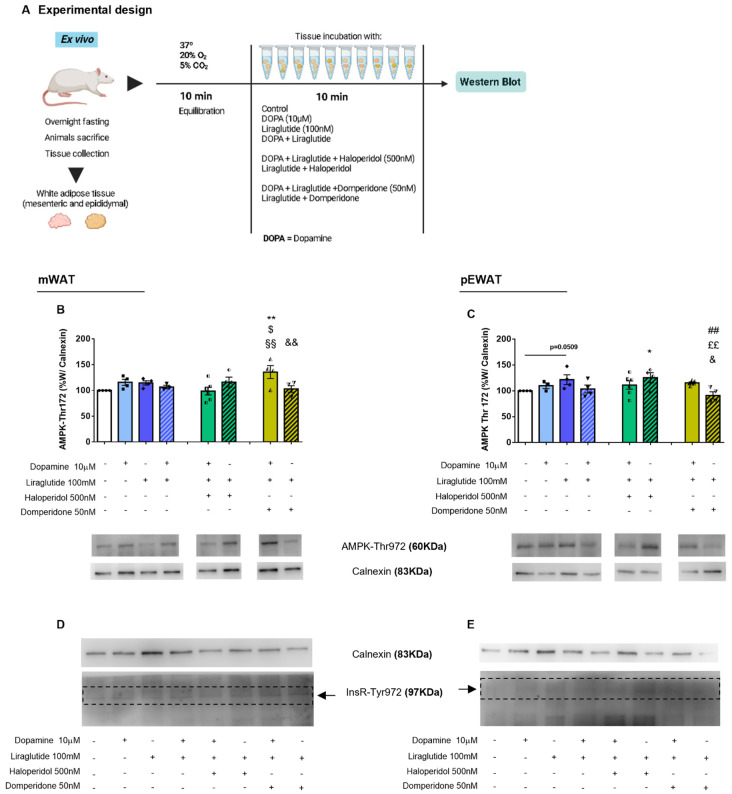
The metabolic crosstalk effect of Dopamine – GLP 1 on AMPK activation in white adipose tissue. AMPK-Thr172 and insulin receptor (InsR)-Tyr972 activation were assessed by Western Blot, in both mWAT and pEWAT explants incubated for 10 min with different agonists/antagonists to unravel the metabolic crosstalk between dopamine and GLP-1 (**A**). AMPK-Thr172 activation increased in the presence of domperidone (antagonist of D2R and D3R) co-incubated with dopamine + liraglutide, and this effect was lost in the absence of dopamine (**B**). In pEWAT, liraglutide alone and in combination with haloperidol increased AMPK-Thr172 levels, independently of dopamine co-incubation (**C**). Moreover, in the absence of dopamine, co-incubation of liraglutide + domperidone decreased AMPK-Thr172 activation (**C**). InsR-Tyr972 activation remained unchanged between conditions in both mWAT and pEWAT (**D**,**E**). Bars represent means ± SEM. One-way ANOVA test was performed. * Different from control; # different from liraglutide; $ different from dopamine + liraglutide; § different from dopamine + liraglutide + haloperidol; £ different from liraglutide + haloperidol; & different from dopamine + liraglutide + domperidone. Level of significance: 1 symbol, *p* < 0.05; 2 symbols, *p* < 0.01.

**Figure 5 ijms-24-02464-f005:**
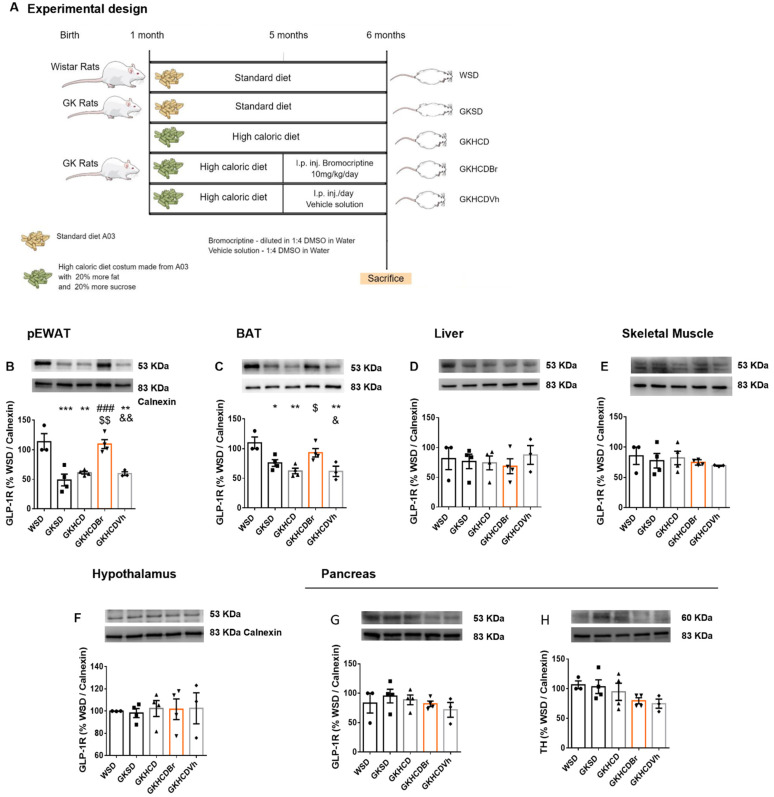
GLP-1 signaling in peripheral tissues after D2R agonist, Bromocriptine treatment in lean and obese type 2 diabetic rats. Type 2 diabetic rats fed a high-caloric diet treated with bromocriptine (experimental design depicted in (**A**)) presented increased levels of GLP1-R in both pEWAT (pEWAT, (**B**)) and brown adipose tissue (BAT, (**C**)) compared with lean and obese type 2 diabetic animals GKSD, GKHCD, and GKHCD_Vh. Bromocriptine treatment did not elicit this effect in the liver (**D**), muscle (**E**), and hypothalamus (**F**), and thus GLP1-R levels were similar between groups. In the pancreas, both GLP1-R and TH levels remained unchanged between groups (**G**,**H**). Bars represent means ± SEM. Differences between groups were assessed by one-way ANOVA test with Tukey post-hoc corrections for multiple comparisons. Symbols represent: * Different from WSD; # different from GKSD; $ different from GKHCD; & different from GKHCD_Br. Level of significance: 1 symbol, *p* < 0.05; 2 symbols, *p* < 0.01; 3 symbols, *p* < 0.001.

**Figure 6 ijms-24-02464-f006:**
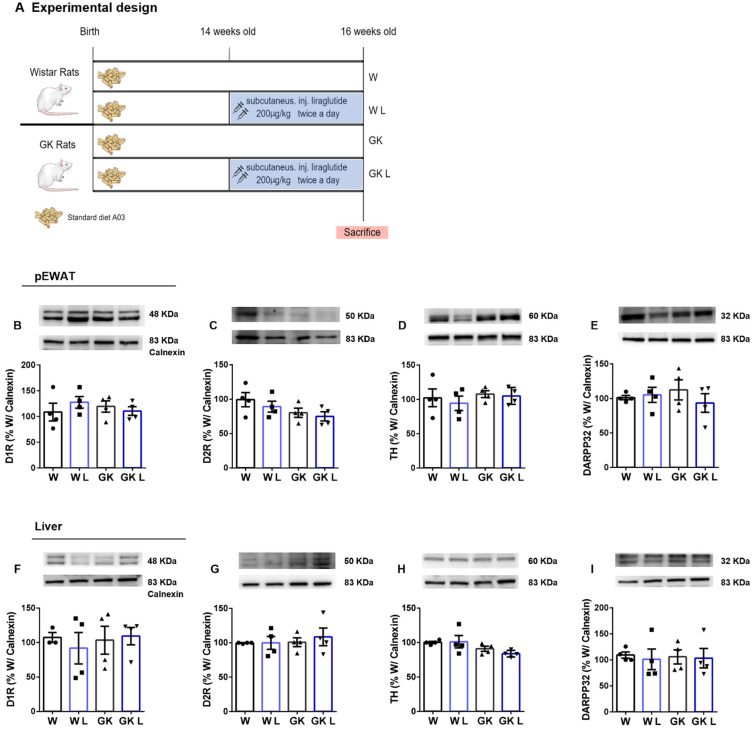
Dopaminergic signaling in peripheral tissues after liraglutide treatment. Liraglutide treatment (experimental design depicted in (**A**)) did not alter D1R (**B**), D2R (**C**), TH (**D**), and DARPP-32 (**E**) levels in pEWAT. Similarly, in the liver, proteins from dopaminergic signaling remain unchanged between groups after liraglutide treatment (**F**,**G**,**H**,**I**). Bars represent means ± SEM. Differences between groups were assessed by one-way ANOVA test with Tukey post-hoc corrections for multiple comparisons.

**Figure 7 ijms-24-02464-f007:**
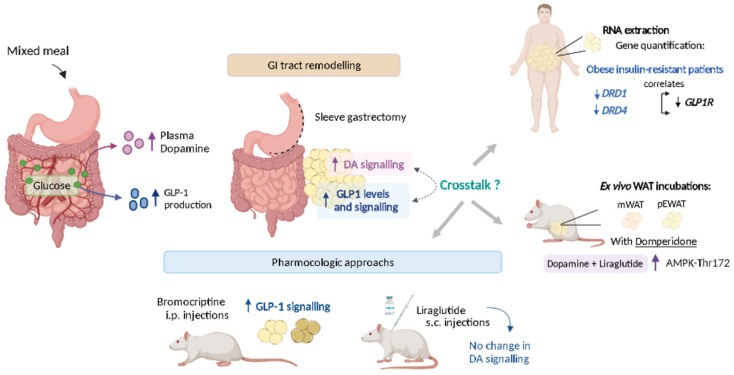
Summary figure of relevant findings. Plasma dopamine levels are postprandially elicited by dietary glucose, and its action in WAT is regulated by the gut. Dopaminergic signaling is correlated to *GLP-1R* in human adipose tissue and their signaling crosstalk regulates the catabolic activity of adipose tissue. Further, dopamine regulates GLP-1R levels in rat adipose tissue, but not the opposite.

## Data Availability

Not applicable.

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
