# Peer review of "Circulating Dopamine Is Regulated by Dietary Glucose and Controls Glucagon-like 1 Peptide Action in White Adipose Tissue"

_ijms, 2023, doi:10.3390/ijms24032464_

Round 1

Author Response

Dear Editors,

We thank the reviewers for the time invested in the revision process and for the constructive comments that resulted in improvements in the manuscript. In this document, we respond, point by point, to the concerns and questions raised. The alterations made in the manuscript following the reviewers’ questions are highlighted in the text (and detailed in each reply) so that they can be easily found.

We believe we have made all the required alterations and hope that they meet your expectations.

Reviewer 1:

Specific concerns:

  1. Major:
  2. According to Mezey et al dopamine in the duodenal juice is probably from the pancreas (Proc Natl Acad Sci USA. 1996;93(19):10377-82).

We have now considered this study and the involvement of the pancreas in peripheral dopamine production (highlighted in yellow - lines 45-47).

  1. There are some concerns with the experimental design. For example, it is not clear why the effect of haloperidol alone is compared with the effect of the combination of dopamine+liraglutide+ domperidone (lines 159-161).

We acknowledge that the particular sentence was not clear enough, thus making it seem that we were comparing haloperidol’s effect alone with dopamine+liraglutide+domperidone. We have rewritten it to improve clarity and avoid misleading interpretation, thank you for calling it to our attention (highlighted in yellow - lines 189-196).

  1. Lines 229-234. The data on GLP1 reactivity are not shown in Figures A1A-A1F.

We apologize for that. As figures are uploaded directly in the manuscript, didn’t submit the Supplementary figure that was included in the pdf file with all the figures. In this figure, you may now find data on GLP-1, DPPIV, and TH protein levels in the ileum of bromocriptine-treated animals (A1A-A1C). Figure A1D comprises GLP-1 immunoreactivity in the ileum and Figures A1E-A1F show the GLP-1 plasma levels of bromocriptine-treated animals after a mixed-meal challenge.

  1. The justification of the doses and concentrations of the drugs used are not given.

Yes, indeed, but we intend to clarify that (in the materials and methods section). In the ex vivo explants experiments (Figure 4) we followed the same design as in our previously published work (Front Pharmacol. 2021 Sep 9;12:713418.). Haloperidol was used in a concentration that blocks both D1R and D2R (Br J Pharmacol . 2015 Jan;172(1):1-23.). Domperidone was used in the same concentration as before for similar experimental conditions (Front Pharmacol. 2021 Sep 9;12:713418.). Liraglutide’s concentration was the same as we have used previously in another study with explants (Pharmacol Res . 2020 Nov;161:105198.).

In experiments with bromocriptine administration (Figure 5 and Supplementary Figure) we chose 10 mg/kg/day as we have previously done this protocol which resulted in substantial improvement in metabolic profile, with no adverse effects on the animals (Mol Metab. 2021 Sep; 51: 101241.). The same rationale behind liraglutide administration (Pharmacol Res . 2020 Nov;161:105198.).

  1. Minor:
  2. Lines 21-22. According to the IUPHAR/BPS Guide to Pharmacology, bromocriptine is not only a D2R and D3R agonist but also 5-HT1D receptor, 5-HT1A receptor, 5-HT7 receptor, and α2A-adrenoceptor agonists (https://www.guidetopharmacology.org/GRAC/ LigandDisplayForward?tab=biology&ligandId=35).

The information regarding bromocriptine’s agonism to the 5-HT receptors and the α2A-adrenoceptor was added (highlighted in yellow - lines 42-44).

  1. Why was a D2R and D3R agonist chosen when D2R immunoreactivity is only found in somatostatin-immunoreactive δ cells but not insulin, glucagon or pancreatic polypeptide immunoreactive cells while no immunoreactivity was seen for D3R or D4R in rat islets? (Cell Tissue Res. 2014;357:597-606)?

We completely understand your point of view. The main reason for choosing bromocriptine is that it is currently an FDA-approved drug for type 2 diabetes. We have been working with bromocriptine for the past few years as we are interested in unravelling how the peripheral dopaminergic system modulates lipid and glucose metabolism in the muscle and liver, but especially in the white adipose tissue, and thus understand the underlying molecular mechanisms for bromocriptine’s beneficial metabolic effects. (Mol Metab. 2021 Sep;51:101241. and Front Pharmacol. 2021 Sep 9;12:713418.). Since it is an approved drugs, our intention is to understand how can it be used as a therapeutic approach for obesity and other metabolic disorders

  1. Line 23. Please state that liraglutide is a GLP-1 receptor agonist.

This has now been added to the abstract (highlighted in yellow).

  1. Lines 25 and 26. Please indicate the reason why dopamine upregulated GLP-1- induced AMPK activity in mesenteric WAT in the presence of the D2R inhibitor domperidone.

As stated in the discussion section (lines 378-396) we believe that dopamine has a synergic effect on liraglutide-mediated AMPK activation. However, this might be mainly happening via D1R, associated with higher catabolic activity. Nonetheless, dopamine has more affinity toward D2R, and thus this incremental effect on liraglutide-mediated AMPK activation is only seen upon D2R blocking, and not on dopamine+liraglutide alone for instance.

  1. Line 27. According to IUPHAR/BPS Guide to Pharmacology, domperidone is D2R and D3R antagonist (https://www.guidetopharmacology.org/GRAC/LigandDisplayForward?tab=biology&li gandId=965).

We thank the reviewer for the clarification and this issue was now solved since we added information regarding domperidone being a D3R antagonist (556-559).

  1. Lines 44-46. The statement that the gastrointestinal tract is the principal source of circulating dopamine in the periphery should have a reference. Soares-da-Silva et al reported intestinal dopamine production in the 1990’s (Gastroenterology. 1994 Sep;107(3):675-9).

Indeed a reference was missing, thank you for providing it. It has now been added.

  1. Lines 78-80. The results in Figure 1C are described before those in Figures 1A and 1B. Is it possible that the high glucose concentrations and therefore their high osmolalities, prevented their absorption? What would have happened if the osmolality of the 60 mg/kg glucose was adjusted to that found in the 200 mg/kg and 500 mg/kg glucose? What were the osmolalities of the different feedings?

Yes, the results in Figure 1C are mentioned before those in 1A and 1B. Nonetheless, they are referred to all in the same sentence, and thus we were expecting to keep it that way, as we believe it improves readability and makes more sense to address the alteration in plasma dopamine levels in the mixed and then verify the absence of difference in the control groups (1A and 1B).

Regarding the osmolalities of the glucose solutions, in fact they might differ, as we kept a constant volume of 2 mL for all of the experimental conditions. Although understanding your concern, we think it was reasonable to maintain the volume at 2 mL for 2 main reasons: 1) the vehicle group received 2 mL of distilled water through gavage to determine if dopamine fluctuations would be attributable to gastric distension and peristaltic movements rather than to the nutritional content. Therefore, if we changed the volumes among the different glucose dosages, we would need to control for that with different vehicle groups according to each volume. 2) the recommended, or even, maximum volume for intragastric delivery in Wistar adult rats is around 2/3 mL (Administration Of Drugs and Experimental Compounds in Mice and Rats | Research Support (bu.edu)), thus increasing this volume to adjust osmolality for the higher doses of glucose would probably be unreasonable.

  1. Lines 117-119 (Figure 2D). There are no alterations in plasma dopamine levels in GKHCD rats. Please interpret this finding.

Besides having type 2 diabetes, these rats become obese when maintained on a high-caloric diet for 6 months. Obesity is described to alter the gut hormones secretion in the post-prandial state either in humans or rodents (Obes Rev . 2021 Feb;22(2):e13130. and Obes Surg . 2015 Jan;25(1):7-18.). Indeed, we see that here for GLP-1 (Supplementary Figure A1F). In healthy Wistars we saw an elevation of plasmatic dopamine after a mixed meal and glucose consumption (Figure 1), which are absent following a mixed meal in GK rats, which we believe to be an intrinsic feature of these rats. GKHCD maintained this phenotype mimicking the phenotype of obesity and type 2 diabetes. This disrupted dopamine’s responsiveness should be also studied in other diabetic models in the future.

  1. Line 146. Do the authors mean Figures 3F, 3G, and 3H instead of 1F, 1G, and 1H?

Yes, it is a typo that was now corrected.

  1. Line 155. Haloperidol is not a D1R antagonist; Ki of haloperidol is 2 nM for D2R and 83 for D1R (ACS Chem Neurosci. 2017 Mar 15;8(3):444-453). The haloperidol pKi for rat D2R is 8.3 and 8.5 for domperidone (https://www.guidetopharmacology.org/GRAC/LigandDisplayForward?tab=biology&li gandId=965)

In line 557, we refer to haloperidol as an antagonist for both D1R and D2R, since it binds both types of dopamine receptors, although with different affinities. Accordingly, with the International Union of Pharmacological societies database for G coupled receptors, pKi of haloperidol for D1 is between 7.6 and 8.2 and for D2 is 8.3.

  1. Figure 4B. There are no studies showing the effect of haloperidol alone or domperidone alone. The concentrations used need to be justified. The incubation time should be indicated in the Figure Legends. Lines 164-165. What is/are the receptor(s) acted upon by dopamine; could it be D1R? Why are the effects of dopamine+liraglutide+domperidone different between mWAT and pEWAT?

The incubation time was now added to the legend of Figure 4. Regarding the concentrations used for haloperidol and domperidone, we selected 500 and 50 nM, respectively, since we have previously used them in ex vivo experiments, similar to this one, with liver, muscle, and adipose tissue Wistar rat explants (Front Pharmacol. 2021 Sep 9;12:713418.).

We strongly believe that upon D2R blocking by domperidone, dopamine’s action is on D1R, since it is coupled to a Gs protein and exerts lipolysis in 3T3-L1 adipocytes via hormone-sensitive lipase activation, thus being associated with a catabolic response, that could explain the rise in AMPK-Thr172 levels. On lines 378-396 (discussion section), we had already given this explanation.

Here, the differences between fat pads were also addressed. The difference in effects upon dopamine+liraglutide+domperidone between mWAT and pEWAT might be due to the differences in tissue metabolism. It is known that white adipose depots are quite different, with the mesenteric being anatomically and metabolically different from the epididymal (Miyata et al. 2016, Sci Rep). In fact, the mesenteric depot is known for having a higher triglyceride turnover rate (Obes Res . 1993 Nov;1(6):459-68. ) draining free fatty acids to the hepatic portal vein and exposing the liver to high concentrations of lipids (e.g. Matsuzawa 2008 Cell Metabolism). Further, the mesenteric adipose tissue might have more sensitivity to dopamine’s action, due to closeness to one of the main producers, the gut.

  1. Figure 5. Why was calnexin chosen as the loading control?

Calnexin is an endoplasmic reticulum-associated protein that suffers less or almost no alterations in its quantity independently of adipocytes' area/volume. Since we have groups of diet-induced obesity, calnexin is considered a better loading control rather than cytoskeleton-associated proteins, such as actin or tubulin, whose quantity will vary according to the adipocytes' area/volume.

  1. Reference 2 does not include the name of the journal, volume, and page numbers (Minerva Endocrinol. 2016, 41(1), 43-56).
  2. Reference 9 does not include the volume of the journal “Mol Metab. 2021, 51, 101241.
  3. There are several errors in syntax and grammar. For example, Line 23: “In contrast” should be “By contrast” (https://english.stackexchange.com/questions/7642/what-isthe-difference-between-by-contrast-and-in-contrast). Line 74. “Disclose” should be changed to “determine”. Lines 75-76. “10-week-old” may be better than “10-weeksold”.

An effort was made to correct all typos, syntax/grammar errors and references.

Reviewer 2 Report

The main objective of this study was to examine the role of peripheral dopamine in metabolic regulation by focusing on nutritional cues that may regulate its release and potential reciprocal interactions between dopamine and GLP-1 in metabolic organs such as liver and adipose tissues.  The present data complement previously published papers from this group and add important information to the overall knowledge on these topics.  In general, the studies with the different rat strains and their treatments are justified and well conducted.  On the other hand, the results of the study with human tissues are far from convincing and do not add significantly to an otherwise good and important investigation. 

Specific points:

11. Delete the section on human studies. The data in Fig 3 are much too spread and the presumed correlation between GLP1R and dopamine receptors among the different groups of patients is not apparent.  I assume that these were stored, rather than freshly collected human tissues, because otherwise, studies could have been conducted with incubated tissues subjected to various treatments.

22. It is incorrect and oversimplified to assume that most peripheral dopamine originates from the gut.  In fact, multiple sources contribute to circulating dopamine, including the many tissues that express TH (adrenals, skin, adipose tissue, lymph nodes, sympathetic nerve endings, etc), as well as the significant conversion of L-DOPA (from both internal and external sources), to dopamine. 

33. Without exception, the lettering in all figures (especially the symbols for significance) are much too small and can hardly be seen even with a magnifying glass.  All figures must be redrawn while paying special attention to their final size in the actual print. 

44. Fig 1: Analysis of plasma dopamine at only 4 time points does not establish consistent periodic fluctuations.  

55. Fig 4: The effects of dopamine on AMPK in the two types of adipose tissue may be statistically significant but are very small and unimpressive.

66. References: Many are incomplete (i.e.: 2,7,10, 16, ,20, 22, 24).  Please carefully insert the missing information in all references.   

Author Response

Dear Editors,

We thank the reviewers for the time invested in the revision process and for the constructive comments that resulted in improvements in the manuscript. In this document, we respond, point by point, to the concerns and questions raised. The alterations made in the manuscript following the reviewers’ questions are highlighted in the text (and detailed in each reply) so that they can be easily found.

We believe we have made all the required alterations and hope that they meet your expectations.

Reviewer 2:

  1. Delete the section on human studies. The data in Fig 3 are much too spread and the presumed correlation between GLP1R and dopamine receptors among the different groups of patients is not apparent. I assume that these were stored, rather than freshly collected human tissues, because otherwise, studies could have been conducted with incubated tissues subjected to various treatments.

In fact, the human visceral adipose tissue samples were stored at - 80 C, they were not freshly collected samples, and thus a similar experiment as the one done with the Wistar rat ex vivo explants was not possible to conduct in this case. Performing ex vivo experiments in human tissue is extremely difficult because different results between patients may be due to real differences between them, or just differences in the experimental conditions over time. This is particularly relevant since usually samples from controls and patients are not collected at the same time or even on the same day. Not having internal controls in all experiments is extremely dangerous to produce biased results. Despite understanding your concern, we aim to maintain Figure 3, as it supports the interplay between GLP-1 and dopamine receptors. In agreement with that, the other appointed reviewer, highlighted the importance of having the human studies with the receptors’ expression on adipose tissue, since this supported the evidence gathered from the animal studies.

  1. It is incorrect and oversimplified to assume that most peripheral dopamine originates from the gut. In fact, multiple sources contribute to circulating dopamine, including the many tissues that express TH (adrenals, skin, adipose tissue, lymph nodes, sympathetic nerve endings, etc), as well as the significant conversion of L-DOPA (from both internal and external sources), to dopamine.

The other primary producers of peripheral dopamine were now mentioned in the introduction section (highlighted in yellow - lines 45-47).

  1. Without exception, the lettering in all figures (especially the symbols for significance) are much too small and can hardly be seen even with a magnifying glass. All figures must be redrawn while paying particular attention to their final size in the actual print.

Figures have been redrawn and resized to increase the reading, especially the symbols for significance.

  1. Fig 1: Analysis of plasma dopamine at only 4 time points does not establish consistent periodic fluctuations.

Although we understand your concern, the amount of blood we can collect from each rat only allows us to study plasma dopamine fluctuations at 4-time points.

  1. Fig 4: The effects of dopamine on AMPK in the two types of adipose tissue may be statistically significant but are very small and unimpressive.

Indeed, in both the graphs and the chosen representative WB images, the differences between groups came across appear to be very mild. In the graphs that may be due to a large number of bars and thus the scale of the graph camouflages the differences. Nonetheless, the mean values have around 20-35% increase in the statistically significant differences (dopamine+liraglutide+domperidone in mWAT; liraglutide in pEWAT and liraglutide+haloperidol in pEWAT).

  1. References: Many are incomplete (i.e.: 2,7,10, 16, ,20, 22, 24). Please carefully insert the missing information in all references.

All references were corrected

Round 2

Reviewer 1 Report

The authors have satisfactorily responded to this reviewer's concerns.The authors have satisfactorily responded to this reviewer's concerns.

Reviewer 2 Report

No further objections.